# Effect of Ti Content on the Microstructure and Properties of CoCrFeNiMnTi_x_ High Entropy Alloy

**DOI:** 10.3390/e24020241

**Published:** 2022-02-04

**Authors:** Yuhua Chen, Wenkuo Liu, Hongwei Wang, Jilin Xie, Timing Zhang, Limeng Yin, Yongde Huang

**Affiliations:** 1Jiangxi Key Laboratory of Forming and Joining Technology for Aerospace Components, Nanchang Hangkong University, Nanchang 330063, China; 1903085204117@stu.nchu.edu.cn (W.L.); 2003080503103@stu.nchu.edu.cn (H.W.); zhangtm@nchu.edu.cn (T.Z.); huangydhm@nchu.edu.cn (Y.H.); 2School of Metallurgy and Materials Engineering, Chongqing University of Science and Technology, Chongqing 401331, China; yeenlm@163.com; 3State Key Laboratory of Advanced Welding and Joining, Harbin Institute of Technology, Harbin 150001, China

**Keywords:** high entropy alloy, Ti element modification, microstructure, properties

## Abstract

The aim of this study was to investigate the effects of the Ti element addition on the microstructure and properties of CoCrFeNiMn high entropy alloys. The Ti element modified CoCrFeNiMnTi_x_ high entropy alloys were prepared by vacuum arc melting processing. The Ti rich body-centered cubic structure phase was observed in CoCrFeNiMnTi_0.25_ and CoCrFeNiMnTi_0.55_ instead of a simple face-centered cubic structure in CoCrFeNiMn. The amount of the Ti-rich phase depicted an increasing trend with increasing Ti content. Simultaneously, the mechanical properties of CoCrFeNiMnTi_x_ were obviously improved. When the Ti content is 0, 0.25 and 0.55, the microhardness is 175 HV, 253 HV and 646 HV, which has an obvious increasing trend, while the ductility decreased. The tensile properties show a trend of first strengthening and then decreasing, changing from 461 MPa to 631 MPa and then to 287 MPa. When x was 0.55, the solid–liquid transition temperature of the alloy decreased, and the melting temperature range increased.

## 1. Introduction

The design concept of a multiprincipal element high entropy alloy (HEA) was first proposed by Taiwanese scholar Jien-wei Yeh on the basis of his research on amorphous alloys in 1995. They were named high entropy alloys in 2004 [1]. Traditional alloys consist of one or two main elements, but since 2004, high entropy alloys (HEAs) consisting of many main elements have attracted great attention. The HEA is composed of 5 to 13 elements in equal molar ratios or near molar ratios; the atomic percentage of each element is between 5% and 35%, and the entropy value is greater than 1.61 R. The characteristic of HEA is the result of all elements, because the content of each element is less than 50%. Compared with traditional alloys, HEAs have good properties, such as higher hardness [2,3], higher strength [4,5], better wear resistance [6,7,8], corrosion resistance [9,10], high temperature oxidation resistance [11,12] and electromagnetism.

In recent years, the composition control of HEAs has been a focus of research. The microstructure and properties of the HEA can be affected by changing the composition of the alloy and the content of certain elements. For example, with the addition of Nb to the AlCoCrFeNi HEA, the ordered Laves phase is formed, and the compressive yield strength and Vickers hardness increase with increasing Nb content [13]. When Si was added to the alloy, the nanoscale structure was precipitated in the alloy. With increasing Si content, new phases were observed in the grain boundaries, and the alloy changed from plastic fracture to brittle fracture with increasing strength [14]. According to the current research, the addition of Al and Cr increases the hardness of HEAs [15,16]. With the addition of Zr and Cr elements, the yield strength of HEAs continues to increase, but the ductility tends to decrease [16,17]. However, the same element has different effects on different alloys. With increasing Mg content, the hardness of the AlFeCuCrMg_x_ alloy first increases and then decreases [18], while the hardness of the Mg_x_(AlCuMnZn)_100−x_ alloy only shows a decreasing trend [19]. Therefore, it is complex to control the microstructure and properties of HEAs by element content, but it is of great significance.

HEA CoCrFeNiMn is called Cantor alloy [20]. This alloy has excellent ductility, high strength [21,22] and even superplasticity [23]. Moreover, it has high tensile elongation at low temperature [5], and excellent fracture toughness [24], making it a candidate material for low-temperature structural materials. However, the yield strength, hardness and wear resistance of the alloy need to be improved, which affects its further application. It is known that element composition can change the properties of alloys [25]. In recent years, there have been many studies on the composition regulation of Cantor alloy. The addition of Al will sharply increase the strength of Cantor alloys, but the ductility will decrease [26]. Mo element is beneficial to the formation of σ phase in the alloy and increases the compression resistance yield strength [27]. The addition of Sn increases the hardness and strength of as-cast alloy [28]. Nd also has a similar effect on the rolled alloy [29]. Studies have shown that the addition of Ti can not only affect the grain size of the matrix and the precipitated phase and the volume fraction of the precipitated phase, but also significantly change the ductility and strength of the HEA [30,31]. Furthermore, Ti can also enhance the corrosion resistance of HEAs [32]. Therefore, Ti is added to the Cantor alloy, which is expected to obtain ideal properties. This article changes the element composition on the basis of a CoCrFeNiMn alloy by adding different contents of Ti to study the changes in its microstructure and properties.

## 2. Experiment Procedures

CoCrFeNiMnTi_x_ (x = 0, 0.25, 0.55) HEAs were prepared from pure metal bulk materials such as Co, Cr, Fe, Ni, Mn and Ti, where x is the molar ratio of Ti to other elements. For convenience, the HEAs studied in this paper are abbreviated as Ti_(x=0)_ alloy, Ti_(x=0.25)_ alloy and Ti_(x=0.55)_ alloy. A cylindrical CoCrFeNiMnTi_x_ HEA ingot was prepared by VIM-5 vacuum arc melting equipment. The melting was repeated 3 times to ensure the uniform chemical composition of the alloy. Then, the ingot was processed by wire cutting into a sample with a size of 5 mm × 10 mm × 10 mm, cleaned by ultrasonic cleaning in an acetone environment, and then ground and polished. The corrosion solution prepared with 5 g FeCl_3_ + 5 mL HCl + 50 mL C_2_H_5_OH was treated, and then microstructure observation, phase analysis and hardness tests were carried out.The dimensions of the tensile test pattern are shown in Figure 9. Approximately 10 mg of metal chips was cut off from the ingot for differential scanning calorimetry.

The metallographic morphology was observed by Chinese Jiangnan brand MR5000 inverted metallographic microscope. The microstructure and composition were analysed by Hitachi SU1510 scanning electron microscope (SEM), equipped with Oxford INCAx-act energy spectrometer. The phase analysis was carried out by using German Brooke D8ADVANCE-A25 X-ray diffractometer with copper Kα radiation using a scanning step of 0.02° and a scanning speed of 2°/min. The FEI Talos F200X field emission transmission electron microscope (TEM) was used to analyse the fine microstructure, and the transmission pattern was prepared by an ion-beam tinning device. Moreover, The Qness Q10 A+ type high-precision microhardness tester was used for hardness testing; the loading force was 500 g, and the pressure holding time was 15 s. A MTS Exceed 40 series microcomputer controlled electronic universal testing machine was used for tensile test. The German Netzsch 404 F3 differential scanning calorimeter was used for the DSC test.

## 3. Results and Discussion

### 3.1. Microstructure Analysis

The microstructures of CoCrFeNiMnTi_x_ HEAs with different Ti contents are shown in Figure 1. Figure 1a–c show the microstructure of the CoCrFeNiMnTi_x_ HEAs, and Figure 1d–i show the microstructure morphology under scanning electron microscopy. Figure 1a shows HEA that does not contain Ti, that is Ti_(x=0)_ alloy. A small number of pores were unevenly distributed in the casting structure, and the pore diameter was less than 5 μm; for the structure containing Ti, no obvious pores were observed, as shown in Figure 1b,c. The dendritic structure was observed in the Ti_(x=0)_ alloy, and the composition analysis was performed at the position marked in Figure 1g. The results are shown in Table 1. The enrichment of Mn and Ni elements was found in the dendritic-like structure, while the interdendritic area was depleted of these elements. This also confirms the findings of related scholars [33].

When Ti was added to the HEA, as shown in Figure 1b,e,h, a new precipitation phase was found in the matrix structure at x = 0.25. The precipitated phases are uniformly distributed in the structure and present an irregular island structure. The results of the energy spectrum analysis at the position marked in Figure 1h are shown in Table 1. There was an obvious segregation phenomenon. The Ti content was higher in the precipitated phase and lower in the matrix phase. Therefore, the precipitated phase is a Ti-rich phase. In addition, the matrix structure was rich in Cr and Fe elements, while these elements were depleted in the Ti-rich phase. The Map scan results of CoCrFeNiMnTi_0.25_ HEA in Figure 1e are shown in Figure 2. The phenomenon of enrichment of Ti and depletion of Cr and Fe in the precipitate can be seen more intuitively. When Ti element is added to the base alloy, new phases rich in Ti will appear, while Cr and Fe elements are enriched slowly, so most of them are retained in the base microstructure.

With the increase of Ti content, the island structure of the Ti-rich phase tends to expand and increase, as shown in Figure 1c,f,i. When x = 0.55, the island tissue dispersed in the matrix tissue with the size increased, and some parts were connected together. In the field of view observed by scanning electron microscopy, the area of island-like tissue covers more than half of the field of vision, as shown in Figure 1f. The energy spectrum analysis shows that there is still enrichment of Ti elements in the Ti-rich phase, while the distribution of other elements is relatively uniform, especially Cr and Fe. The previous obvious segregation phenomenon is weakened. 

When no other elements are added to the CoCrFeNiMn HEA, the structure is a single structure without precipitation phases [24]. When Ti was added, the micro-pores appearing during casting disappeared, and a Ti-rich phase appeared. The Ti-rich phase content in the alloy is 9.28% at x = 0.25 but the content of the Ti-rich phase increases to 60.66% at x = 0.55, which is approximately 6.5 times that of the former. Comparing the microstructure of Ti_(x=0.25)_ alloy and Ti_(x=0.55)_ alloy, it can be found that with the increase of Ti element, the Ti-rich phase grows up and is dispersed in the base layer structure. Moreover, the segregation of other elements improves.

### 3.2. X-ray Diffraction and TEM Analysis

The XRD patterns of the three CoCrFeNiMnTi_x_ HEAs studied in this paper are shown in Figure 3. When x = 0, there are obvious diffraction peaks at 43.68°, 50.68° and 74.63° and the corresponding crystal orientations are {111},{200} and {220}, respectively. CoCrFeNiMn alloy is a single FCC structure. According to PDF card analysis, this structure is similar to (Fe.Ni) solid solution. The other three elements were substituted at the positions of Fe and Ni atoms. When x = 0.55, in addition to the above three diffraction peaks, there are four groups of diffraction peaks at 43.25°, 45.68°, 48.10°, 52.55° and 79.09° in the XRD pattern. The corresponding crystal orientations are {332}, {422}, {431}, {440} and {741}. According to the PDF card, the newly formed phase is a BCC structure. Analysis of Figure 1i and Table 1 shows that the newly formed phase was rich in Ti, which was called the Ti-rich phase. When x = 0.25, except for the diffraction peak of FCC(Fe.Ni) structure, the diffraction peak of BCC (Ti-rich phase) was weak. According to Figure 1b,e, the content of the precipitated phase (Ti-rich phase) in Ti_(x=0.25)_ alloy was lower and the distribution was more dispersed. Therefore, in the XRD pattern, the intensity of the diffraction peak is weak. 

The CoCrFeNiMn HEA has a single FCC (Fe.Ni) structure. With the addition of Ti, a new BCC (Ti-rich phase) structure was formed on the matrix structure. Moreover, with the increase of Ti, the BCC structure tends to increase obviously. In this process, the crystal structure of the alloy changes from FCC to (FCC + BCC). Ti_(x=0)_ alloy is a single FCC structure, while for Ti_(x=0.25;X=0.55)_, the two alloys coexist with FCC and BCC.

To further prove the above conclusion, TEM analysis was performed on Ti_(x=0.55)_ alloy. Figure 4 demonstrates TEM bright-field images of Ti_(x=0.55)_ alloy in the as-solidified condition. The matrix phase and Ti-rich phase existed in the Ti_(x=0.55)_ alloy. The matrix phase is composed of (Fe.Ni) solid solution phase with FCC structure, and the crystal structure of Ti-rich phase corresponds to Cr_0.08_Ti_0.92_, which is composed of Ti-containing solid solution with BCC structure. In the CoCrFeNiMn system HEA, the addition of Ti promotes the formation of the BCC secondary phase. In addition, the element map scan was carried out under the TEM field of view, and the results are shown in Figure 5. A small amount of Ti compound particles with a size of 1–3 μm were found at the interface of the two phases. After the component test, the material may be titanium oxide. With the addition of Ti, the BCC solid solution phase begins to precipitate, some areas reach the precipitation saturation value, and Ti appears in the form of compounds.

### 3.3. Analysis of Solid–Liquid Transition Temperature

The DSC curves of CoCrFeNiMnTi_x_ HEA with different Ti contents are shown in Figure 6. Obvious endothermic peaks were found in the heating process of the three alloys. There is one endothermic peak in Figure 6a,b, while two endothermic peaks appear in Figure 6c with partial overlap. The reason is that in Ti_(x=0.55)_ alloy, in addition to the (Fe.Ni) phase with an FCC structure, there are also Ti-rich phases with a BCC structure. During the solid–liquid transition, the Ti-rich phase melts first, and the (Fe.Ni) phase also melts before the end of the process. Therefore, two endothermic peaks that partially overlap are found in the DSC curve. There is only a single (Fe.Ni) phase in Ti_(x=0)_ alloy. Although the Ti_(x=0.25)_ alloy has a Ti-rich phase, the content is relatively small. The matrix phase plays a decisive role, and there is no second endothermic peak. However, its solid–liquid transition temperature is reduced. The solid–liquid transition temperature range of Ti_(x=0.55)_ alloy becomes larger than that of the other two alloys. Ti_(x=0.55)_ alloy is 132 °C, while the other two phases are both 39 °C.

Compared with the curves in Figure 6a–c, the solid–liquid transition temperature of the alloy moves backwards (that is, the solid–liquid transition temperature decreases) with the addition of Ti. Ti_(x=0)_ alloy starts melting at a temperature of 1299 °C; the melting end temperature (also known as liquidus temperature) is 1338 °C. The liquidus temperature of Ti_(x=0.25)_ alloy is 1277 °C, which is 61 °C lower than the former. The liquidus temperature of Ti_(x=0.55)_ alloy is 1215 °C, which is 62 °C lower than that of Ti_(x=0.25)_ alloy. Therefore, the solid–liquid transition temperature decreases with the addition of Ti.

### 3.4. Micro Vickers Hardness Analysis

Figure 7 shows the Vickers hardness of the CoCrFeNiMnTi_x_ high entropy alloy. When x = 0, that is, there is no Ti element in the alloy, the hardness of the alloy is 175 ± 20 HV, which is close to the present study [34]. The hardness value of the alloy increases due to the addition of Ti, and with increasing Ti content, the hardness changes obviously. When x = 0.25, the hardness of the alloy increases to 253 ± 10 HV. When x = 0.55, the hardness of the alloy increases to 646 ± 50 HV, and its hardness is significantly higher than that of the former two (x = 0 and 0.25). Some scholars believe that the increase in hardness is due to the increase in lattice parameters of FCC structure and lattice distortion caused by the addition of large Ti atoms to the alloy [35].

In comparison, when the alloy is a single FCC (Fe.Ni) phase, i.e., at x = 0, the hardness value is lowest. The increase in the hardness of the material benefits from the BCC (Ti-rich phase) structure. When BCC (Ti-rich phase) precipitated in the alloy, the hardness increased. Compared with the alloy without Ti (as for x = 0), the hardness change of Ti_(x=0.55)_ alloy is more obvious, almost increased to 4 times, but the hardness increase of the Ti_(x=0.25)_ alloy is smaller. According to the microstructures of the three alloys, as shown in Figure 1a–c and the XRD results, Ti_(x=0.25)_ alloy contains less BCC phase, which has an impact on the overall hardness increase, but the effect is relatively small. The hardness of the material is greatly improved. When the Ti content is increased to x = 0.55, the content of BCC phase increases.

### 3.5. Tensile Performance Analysis

In this study, the tensile properties of the samples were tested. Table 2 shows the test results of the tensile test. The engineering stress–strain curve is shown in Figure 8. Compared with Ti_(x=0)_ alloy, with the addition of Ti element, the strength of Ti_(x=0.25)_ alloy is significantly improved, the tensile yield strength is increased from 212 MPa to 338 MPa, the tensile strength is increased from 461 MPa to 631 MPa, and the tensile strength of Ti_(x=0.25)_ alloy is 36% higher than that of the alloy without Ti element. With the continuous addition of Ti, the strength of the Ti_(x=0.55)_ alloy decreases sharply, and the yield strength and tensile strength are 176 MPa and 287 MPa, respectively, which is lower than that of the alloy without Ti. The tensile fracture surfaces of three samples were observed. Obvious dimples were found at the fracture surface of the Ti_(x=0)_ alloy, which is a typical plastic fracture mode, as shown in Figure 9a,d. No dimples were found at the fracture of Ti_(x=0.25)_ alloy, and there was a uniform fracture section, as shown in Figure 9b,e. Obvious cracks were found in the Ti_(x=0.55)_ alloy, as shown in Figure 9c, and granular materials were found at the fracture of the alloy. According to the composition analysis, the substance is a compound containing Ti, which is close to the compound composition found in the TEM field of vision. The enlarged diagram of the compound is shown in Figure 9h. Ti-rich hard particles can be considered cracking sources that favored the propagation of cracks during deformation. According to the fracture morphology, the two alloys with Ti show a brittle fracture mode.

The addition of a certain amount of Ti can improve the tensile strength of alloy materials, such as Ti_(x=0.25)_ alloys. However too much Ti will reduce the strength of the alloy, such as Ti_(x=0.55)_ alloy. The atomic radius of Ti is larger than that of other elements, which affects the crystal distortion and stability of the FCC solid solution structure. The second phase formed by adding Ti is a Ti-rich phase, which has a BCC structure. The formation of the second phase improves the tensile strength of the material, but reduces the elongation of the material. With the continuous addition of Ti, the Ti-rich phase continues to increase, which theoretically promotes the further strengthening of the tensile strength of the material, but the tensile test shows that the Ti_(x=0.55)_ alloy exhibits a weaker tensile strength. Ti compound was found in the tensile fracture of the alloy, and the same substance was found in the TEM analysis. During the tensile process, the compound particles will form stress concentrations. As the tensile continues, the crack will expand from this place, resulting in unstable fracture. The Ti_(x=0.55)_ alloy exhibits weak tensile strength. Therefore, the addition of Ti can reduce the ductility of the alloy and improve the tensile strength of the alloy, but it is necessary to control the amount of Ti; otherwise, the tensile properties of the material will deteriorate.

## 4. Conclusions

In this paper, the microstructure and properties of CoCrFeNiMnTi_x_(x = 0; 0.25; 0.55) high-entropy alloys with different Ti contents were studied, and the following conclusions are drawn:(1)CoCrFeNiMn alloy is a single FCC (Fe.Ni) phase. After adding Ti, the BCC(Ti-rich phase) structure was found. With the increase of Ti content, the Ti-rich phase depicted an increasing trend. When the Ti content is 0.55, the alloy has a structure in which FCC and BCC coexist, and Ti compound particles appeared in some areas.(2)The solid–liquid transition temperatures of Ti_(x=0;0.25;0.55)_ alloys are 1299 °C–1338 °C, 1238 °C–1277 °C, and 1083 °C–1215 °C, respectively. The solid–liquid transition temperature of the alloy decreased and the melting temperature range extended due to the addition of Ti.(3)With the addition of Ti element, the hardness of Ti_(x=0;0.25 and 0.55)_ alloy was 175 HV, 253 HV, and 646 HV, respectively. The results show that the hardness of the alloy are obviously improved by Ti.(4)The tensile strength of the alloy is increased from 461 MPa to 631 MPa, along with Ti_(x=0)_ alloy to Ti_(x=0.25)_ alloy, but the strength of Ti_(x=0.25)_ alloy is reduced to 287 MPa. A certain amount of Ti will improve the tensile strength of the alloy. The addition of excessive Ti will lead to the production of compounds and reduce the tensile strength of the alloy.

## Figures and Tables

**Figure 1 entropy-24-00241-f001:**
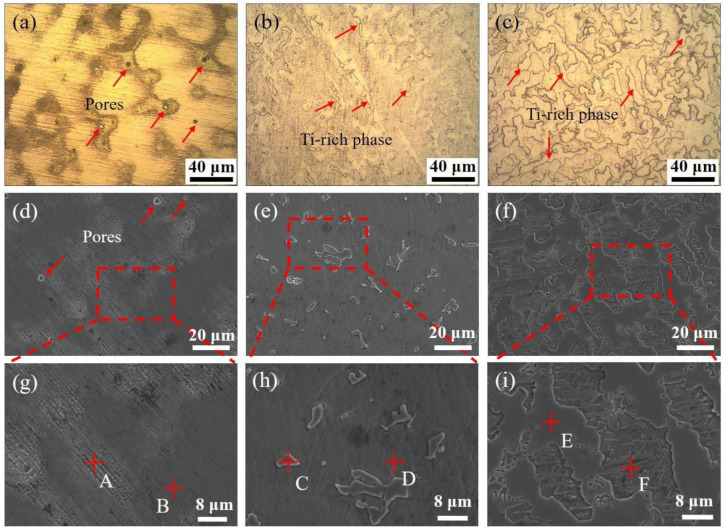
Microstructure of CoCrFeNiMnTi_x_ HEA with different Ti contents: (**a**,**d**,**g**) Ti_(x=0)_ alloy; (**b**,**e**,**h**) Ti_(x=0.25)_ alloy; (**c**,**f**,**i**) Ti_(x=0.55)_ alloy.

**Figure 2 entropy-24-00241-f002:**
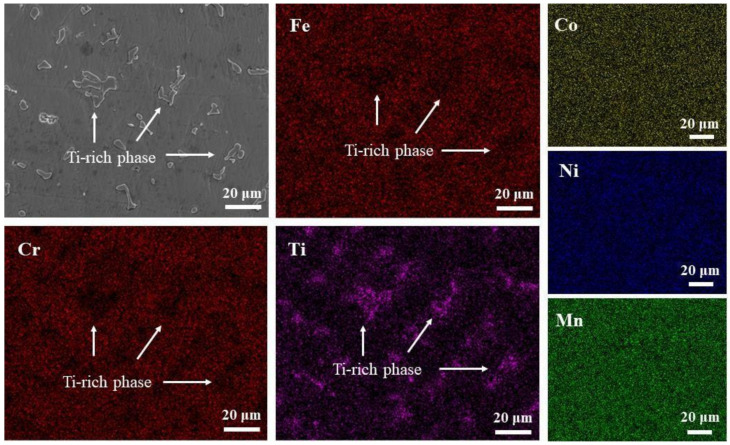
The Map scan results of CoCrFeNiMnTi_0.25_ HEA in Figure 1e.

**Figure 3 entropy-24-00241-f003:**
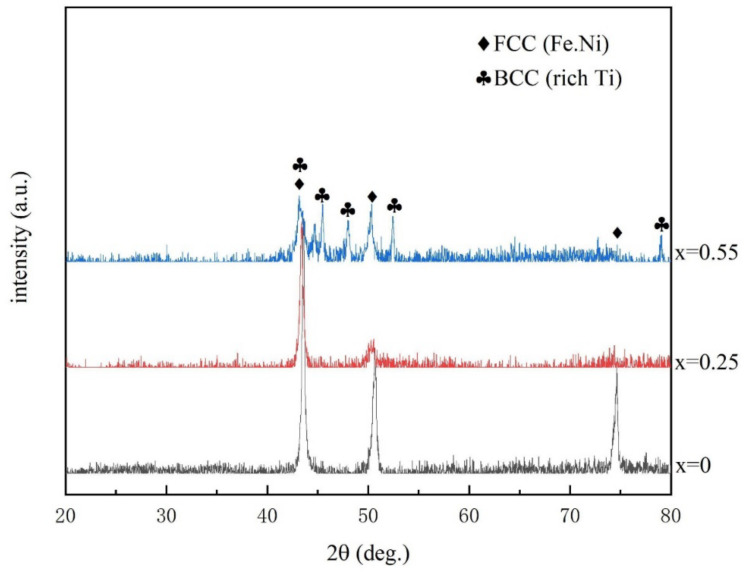
XRD results of the CoCrFeNiMnTix high entropy alloys.

**Figure 4 entropy-24-00241-f004:**
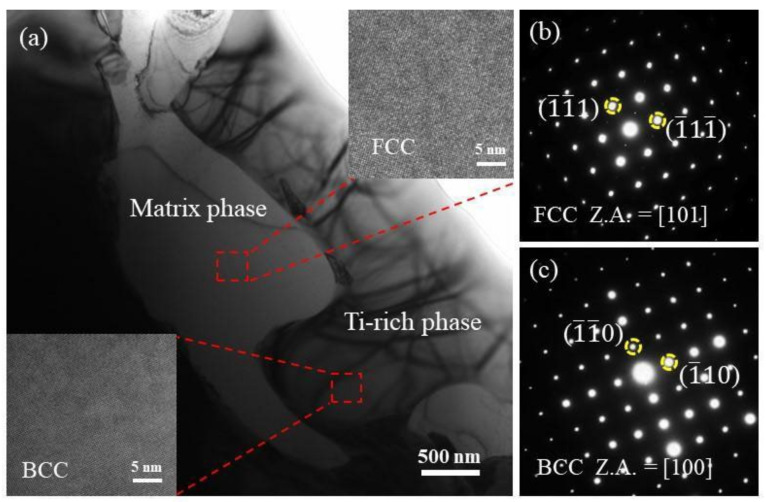
TEM bright-field image of CoCrFeNiMnTi_0.55_ HEA in the as-solidified condition and corresponding selected area electron diffraction (SAED): (**a**) TEM bright-field image; (**b**) SAED of matrix phase; (**c**) SAED of Ti-rich phase.

**Figure 5 entropy-24-00241-f005:**
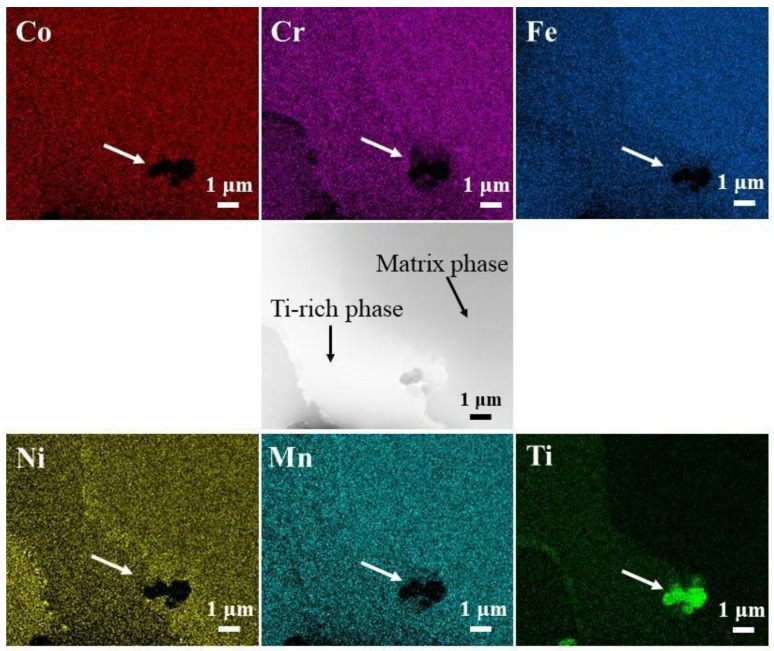
The map scan results of CoCrFeNiMnTi0.55 HEA in HAADF of TEM.

**Figure 6 entropy-24-00241-f006:**
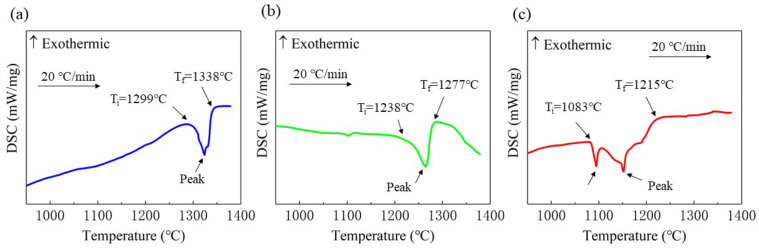
DSC curve of the CoCrFeNiMnTix alloys: (**a**) Ti_(x=0)_ alloy; (**b**) Ti_(x=0.25)_ alloy; (**c**) Ti_(x=0.55)_ alloy.

**Figure 7 entropy-24-00241-f007:**
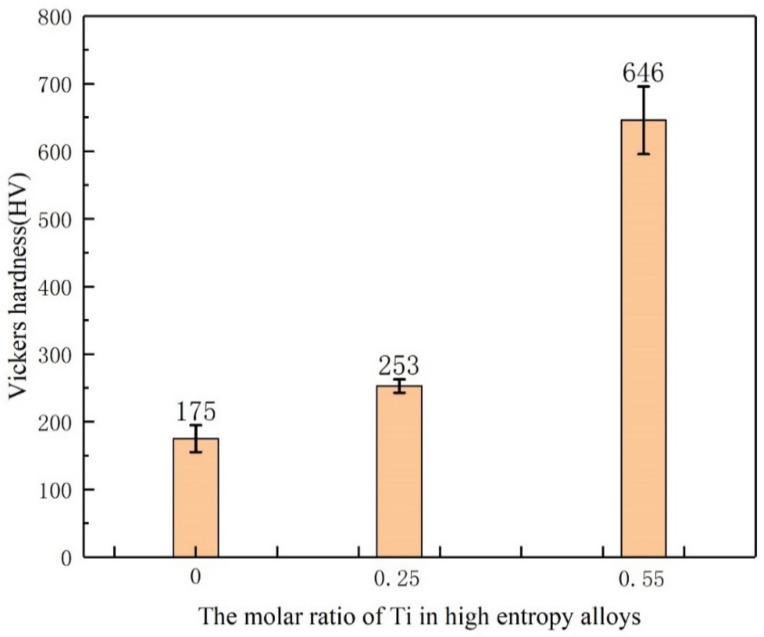
Vickers hardness of the CoCrFeNiMnTi_x_ HEA.

**Figure 8 entropy-24-00241-f008:**
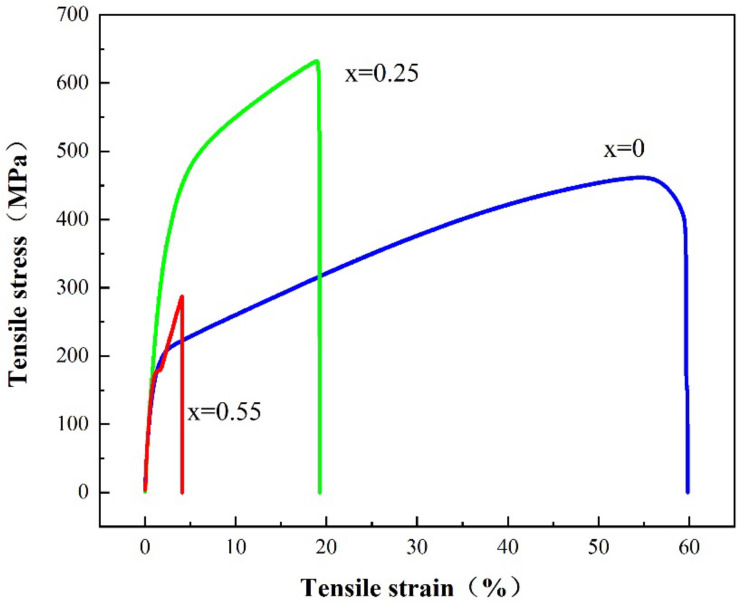
Tensile stress–strain curves of the CoCrFeNiMnTix HEA.

**Figure 9 entropy-24-00241-f009:**
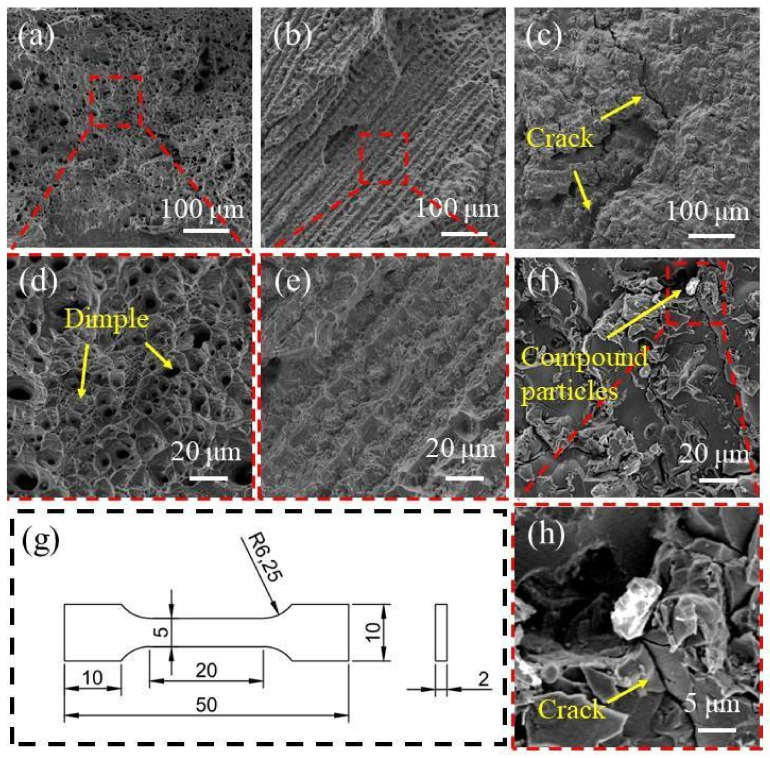
Tensile fracture morphology: (**a**,**d**) Ti_(x=0)_ alloy; (**b**,**e**) Ti_(x=0.25)_ alloy; (**c**,**f**,**h**) Ti_(x=0.55)_ alloy; (**g**) Dimensions of tensile test specimens (mm).

**Table 1 entropy-24-00241-t001:** Chemical composition analysis of each point in Figure 1g–i (at%).

Collection Point	Co	Cr	Fe	Ni	Mn	Ti
A	17.34	16.83	15.22	22.30	28.30	
B	21.41	22.41	22.86	17.65	15.67	
C	19.55	9.99	12.04	21.37	16.01	21.04
D	19.44	21.78	21.24	16.72	17.88	2.93
E	17.64	22.34	18.72	13.11	16.88	11.31
F	17.58	22.11	21.45	15.17	18.48	5.21

**Table 2 entropy-24-00241-t002:** Tensile performance of CoCrFeNiMnTix HEA.

The Molar Ratio of Ti	Tensile Yield Strength (MPa)	Tensile Strength (MPa)	Elongation (%)
x = 0	212	461	55
x = 0.25	338	631	19
x = 0.55	176	287	4

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
