# Peer review of "Effect of Ti Content on the Microstructure and Properties of CoCrFeNiMnTix High Entropy Alloy"

_entropy, 2022, doi:10.3390/e24020241_

Round 1
Reviewer 1 Report
The manuscript entitled: Effect of Ti content on the microstructure and properties of CoCrFeNiMnTix high entropy alloy studies the effect of Ti addition on the Cantor alloy. However, I have the following concerns.
- XRD: The indexed BCC peaks seem to be not correct. For Cu-radiation, the BCC peaks should have the following positions: ~44 - (110), 65 (200), 82.5 (211)... However, the indexed positions here do not match with any BCC nor HCP phase. Attention is needed with peak indexing.
- The calculated compressive yield strengths are not correct. Please double-check with the compressive curves and the table. There seems to be some discrepancy!
- A strong scientific discussion correlating the influence of Ti on the phase formation and structure-property correlation is missing!
- Typos in the manuscript need attention. For instance: Fig. 11 Y-axis caption - Tensile stress (Mpa) should be written as Tensile stress (MPa).
- The English language needs careful revision (probably by a native English speaker)
Reviewer 2 Report
- Original Submission
- Recommendations – Minor revision
- Comments to author
Ref. Entropy - 1547427
Title. Effect of Ti content on the microstructure and properties of CoCrFeNiMnTix high entropy alloy
Overview: The paper presents the Ti effect on the microstructure, mechanical and thermophysical properties of the high entropy CoCrFeNiMn alloy.
It has been found that the addition of Ti in various increasing molar proportions (0.25 and 0.55) increases the hardness of the metal matrix and the compressive strength, decreases the deformability and increases the fragility of the alloy.
The paper is interesting and the conclusions are demonstrated by experiments.
Correction required:
Lines 205-207: Please reformulate. The content is unclear and confusing
The melting temperature range of Ti(x=0.55) alloy is increased, due to the difference of solid-liquid transition temperature between the two phases, Ti(x=0.55) alloy is 132℃, while the other two phases are both 39℃.
The comma must be placed immediately after the previous character, without a space. There are several such typos! (ex. lines 210, 211, 213...)
Please make a correction for the MPa units throughout Chapter 3.5! Including in Fig. 8 and Table 2.
Lines 294-295: The compound particles as the crack source led to the crack propagation.
Suggestion: Ti-rich hard particles can be considered cracking sources that favored the propagation of cracks during deformation.
Conclusions
Line 333: .... melting temperature range increased ...
Suggestion: ... the melting temperature range has been extended
General recommendation: A native English proofreader is required!
Reviewer 3 Report
The scientific research paper “Effect of Ti content on the microstructure and properties of CoCrFeNiMnTix high entropy all” - by Yuhua Chen, Wenkuo Liu, Hongwei Wang , Jilin Xie, Timing Zhang, Limeng Yin, Yongde Huang is a interesting attempt in the field of the Cantor high entropy alloys. In the scientific paper are presented the studies on the effect of Ti content on the properties of HEA with six elements from the CoCrFeNiMnTix system.
Abstract
The content of Ti in the alloy does not increase in the range of 0-55% at., but only 3 chemical compositions are obtained with 0.00, 0.25, respectively 0.55% molar. (Please, correction!!!)
- Introduction
Introduction (32 references), I consider it’s good. The references stop in 2020, and the state-of-the-art is sufficient. HEA alloys that are influenced by the alloying elements must be specified exactly. 2. Experimental procedures
Specify the source of the VIM-5 vacuum melting equipment. I consider that repeating the melting 3 times does not ensure the homogenization of the alloy. I work in my laboratory with at least 5 melts for homogenization. Also specify the origin of the MR5000 inverted metallographic microscope,
3. Results and DiscussionsWhy the TEM were analyzes only on the Ti0.55 alloy?Rows 244, 245, 246 - MPa instead of Mpa. 4. Conclusions The conclusions are clearly formulated based on the research carried out by the authors of the scientific paper. However, small additions and reformulations can be made in the paper, which would provide a better explanation of some of its components. I signal my opinion that I will reproduce below and that the authors must reconsider.
Round 2
Reviewer 1 Report
The authors have addressed most of the concerns raised and the manuscript may now be accepted for publication in the present form.